# Acceptability of intravitreal injections in geographic atrophy: protocol for a mixed-methods pilot study

Jamie Enoch [1], Arevik Ghulakhszian [2], David P Crabb [1], Christiana Dinah [2], Deanna J Taylor [1]

CD and DJT are joint senior authors.

¹Optometry and Visual Sciences, City University of London, London, UK
²Ophthalmology Department, London North West University Healthcare NHS Trust, Central Middlesex Hospital, London, UK

**Correspondence to**
Dr Deanna J Taylor;
Deanna.Taylor.2@city.ac.uk

## ABSTRACT

**Introduction** Age-related macular degeneration (AMD) is a common cause of visual impairment, affecting central vision. Geographic atrophy (GA) is an advanced form of the non-neovascular (dry) type of AMD. Late-stage clinical trials suggest that intravitreal injections of novel therapeutics may slow down the rate of GA progression by up to 30% in 1 year, thus allowing people with GA to preserve central vision for a longer period. While intravitreal injections have become an established treatment modality for neovascular (wet) AMD, it is unknown whether patients with (more gradually progressing) GA would accept regular injections that slow down, but do not stop or reverse, vision loss. Therefore, this mixed-methods pilot study will aim to explore whether regular intravitreal injections will be acceptable as treatment for patients with GA, and the factors that may affect treatment acceptability.

**Methods and analysis** A mixed-methods survey has been designed in collaboration with a GA patient advisory group. The survey comprises of structured questionnaires, semi-structured interview questions regarding patients' perceptions of intravitreal injections and the burden of treatment, and a task eliciting preferences between different potential treatments. Due to COVID-19 restrictions, this study will be conducted remotely by telephone. Thirty individuals will be recruited from NHS Medical Retina clinics at Central Middlesex Hospital, London. Half of the participants will be naïve to intravitreal injections, while half will have previous experience of intravitreal injections for neovascular (wet) AMD. Qualitative data analysis will be conducted using the Framework Method of analysis to identify key themes from participants' accounts.

**Ethics and dissemination** The study received Health Research Authority approval on 23 March 2021 (IRAS Project ID: 287824). Findings will be disseminated through peer-reviewed publications and conference presentations to the medical retina community, as well as through dialogue with patients and macular disease charities.

## Strengths and limitations of this study

► This is the first study to investigate the acceptability of emerging geographic atrophy (GA) treatments for patients living with the condition. The resulting findings will help to understand the psychosocial, demographic and health service factors that may influence treatment acceptability.

► All aspects of the study are benefiting from the input of an advisory group, eight individuals living with GA who are helping to shape the study design, analysis and dissemination.

► Following analysis, findings from this pilot study will be used as the basis for a subsequent quantitative study, where we will recruit a larger sample of patients with GA from a range of sites across the UK.

► Due to the COVID-19 pandemic, this pilot study will be conducted remotely. While we have engaged an advisory group of patients living with GA to help us develop the study in this format, there is a risk of losing non-verbal information during interviews conducted remotely.

► Taking a qualitatively driven mixed-methods approach means that we will be conducting in-depth interviews with a small number of participants from one eye clinic, which limits the generalisability of the study findings.

## INTRODUCTION

Age-related macular degeneration (AMD) is the most common cause of sight loss in developed countries.[1] Sight loss from advanced AMD occurs either as neovascular or 'wet' age-related macular degeneration (nAMD) or geographic atrophy (GA). In the UK, GA is estimated to account for one quarter of legal blindness, and globally approximately 5 million people have GA in at least one eye.[2] Patients with GA generally develop retinal lesions which can lead to irreversible vision loss, and about half of patients develop GA in both eyes within 7 years of initial diagnosis.[3]

There is currently no therapy for GA, although researchers are making significant progress in understanding and developing potential treatments that may slow down the growth of GA lesions. These include oral treatments such as ALK-001 and doxycycline,[4 5] and surgical delivery of gene therapy in phase II trials.[6] Showing promising results and now in late stage clinical trials are drugs

targeting the complement cascade and neuroprotective drugs. These are delivered by injection into the patient's eye, known as intravitreal injections. Table 1 describes intravitreal treatments that have been or are currently being evaluated in phase III clinical trials.

Intravitreal injections are delivered into the patient's eye at regular intervals for an indefinite period, as is currently standard practice for nAMD. The current treatments under investigation are delivered at monthly, bimonthly or 3 monthly intervals and have been shown to slow down GA lesion growth by up to 30% in 1 year, depending on the therapy and how often injections are delivered (table 1).[7] This would translate to lesions taking longer to affect central vision and patients therefore keeping useful central vision for a longer period.

## Learning lessons from treatment of nAMD

Regular intravitreal injections of anti-vascular endothelial growth factor (anti-VEGF) are used extensively in treatment of nAMD to slow or halt disease progression. Anti-VEGF injections have substantially reduced the prevalence of sight loss attributable to nAMD.[8]

However, studies of patients receiving intravitreal injections for nAMD show evidence of significant anxiety, stress, discomfort and pain associated with these injections.[9–12] A recent metasynthesis of qualitative research on experiences of patients with nAMD summarises potential negative impacts of repeated injections:

> The time spent at clinics, the injection itself, overcrowded ophthalmic practices and hospitals, the number of examinations, communication problems with medical staff, information deficits and the side effects of treatment are often debilitating and nerve-wracking for many patients.[13]

Patients also often express concern about the burden on relatives and carers who may transport or accompany them to hospital for these appointments.[10] Indeed, a recent Macular Society survey found that 61.6% of patients receiving anti-VEGF treatment rely on friends or family to take them to and from eye clinic appointments.[14] A complex interaction of patient-related health, demographic, psychological, socioeconomic and geographical factors may combine with characteristics of the treatment delivery and health system and create barriers to adhering and persisting with treatment (see figure 1).[15 16] The nAMD literature suggests that loss to follow-up after initiating treatments is highest in the first year.[16–18] Many determinants of non-adherence/non-persistence in patients treated with intravitreal injections for nAMD are unmodifiable, such as age, visual acuity and ocular or systemic comorbidities when commencing treatment. However, other key barriers—such as distance from the hospital,[17 18] travel times and transportation implications[19]—could be addressed by improving health service delivery, with some small studies suggesting that 'one-stop' style services combining monitoring and injections could improve adherence.[20 21]

Nonetheless, despite high treatment burden, the possibility of further sight loss from nAMD has often been found to outweigh negative experiences and to motivate patients to continue treatment.[10 22 23] In a discrete choice experiment study, desire to prevent deterioration of visual acuity was by far the most significant factor in treatment decision-making, in nearly three-quarters of cases.[24]

| **Table 1** Summary of investigational intravitreal treatments for GA | | |
|---|---|---|
| **Name of drug** | **Trial name (ClinicalTrials.gov identifier)** | **Interim or published results** |
| APL-2 | FILLY (NCT02503332) DERBY and OAKS (NCT03525600 and NCT03525613) | In a phase II trial, APL-2 administered monthly via intravitreal injection showed a 29% statistically significant reduction in the rate of GA lesion growth compared with sham after 12 months of treatment.[36] Two phase III multicentre trials are underway. |
| Zimura (avacincaptad pegol) | GATHER1 phase II/III trial (NCT02686658) GATHER2 phase III trial (NCT04435366) | In a phase II/III trial, avacincaptad pegol administered monthly via intravitreal injection significantly reduced GA growth over 12 months by 27.4% (2 mg cohort) and 27.8% (4 mg cohort) compared with sham.[37] |
| Brimonidine (Brimo DSS) | BEACON (NCT02087085) | In a phase IIb study, Brimo DDS administered every 3 months via intravitreal injection significantly decreased GA growth by 10% at month 24 and by 12% at month 30 compared with sham.[55] |
| Lampalizumab | CHROMA (NCT02247479) SPECTRI (NCT02247531) | No statistically significant improvements. In phase III trials, patients with bilateral GA experienced a consistent decline in visual function over 48 weeks. Measures of visual function were not correlated strongly with GA lesion area.[56 57] |

GA, geographic atrophy.

| Patient-related health and socio-demographic factors | The relationship between the patient and healthcare provider |
|---|---|

**Patient-related health and socio-demographic factors**
- Functional Vision at baseline e.g. best corrected visual acuity (BCVA)
- Age
- Comorbidities (ocular and systemic)
- Distance from hospital
- Caregiver availability
- Transportation options
- Indirect costs of treatment
- Socio-economic status

**The relationship between the patient and healthcare provider**
- Communication between clinicians and patient
- Trust
- Familiarity
- Prior experiences

**Treatment factors**
- Frequency of injections
- Efficacy
- Subjective effectiveness for patient: do they feel it's working?
- Side effects – including pain and discomfort
- Risk of adverse events or complications

**Patient-related sychosocial factors**
- Information
- Motivation
- Self-efficacy
- Health beliefs
- Coping Styles
- State and trait anxiety
- Fear of injections
- Fear of visual impairment

**Health service and system factors**
- Accessibility/geographical proximity of clinic
- Convenience of treatment schedule
- Duration of time spent in clinic
- Satisfaction with service and the process of care

**Figure 1** Summary of factors that may affect acceptability of intravitreal treatments, based on the neovascular age-related macular degeneration literature.[16 58 59]

Even while anxiety about injections may be common, this is often outweighed by anxieties regarding vision loss or the treatment's failure to work effectively.[18]

In contrast to nAMD, where loss of vision is typically sudden and treatment leads to improvements in vision, disease progression and vision loss in GA is a gradual process. Moreover, current intravitreal treatments in late phase development for GA (shown in table 1) slow down the rate of vision loss, but do not stop or reverse vision loss. It is therefore not yet known whether patients with GA will be similarly motivated to adhere to frequent intravitreal treatments, how often they would be willing to undergo them and what factors would make such treatments acceptable to patients with GA. Therefore, it is vital to explore how patients with GA themselves perceive the anticipated benefits and drawbacks of these treatments, and consider their overall acceptability.

### A theoretical framework of acceptability

For the purposes of this protocol and the pilot study, we define acceptability in accordance with Sekhon, Cartwright and Francis' comprehensive review as:

> A multi-faceted construct that reflects the extent to which people delivering or receiving a healthcare intervention consider it to be appropriate, based on anticipated or experienced cognitive and emotional responses to the intervention.[25]

In table 2, we consider the seven components of Sekhon *et al*'s theoretical framework of acceptability, with concrete examples in the context of our study.

### Living with GA: the patient experience

While treatment acceptability for GA is a novel research area, it is also worth noting the very limited research on patient experiences of living with dry AMD and with GA in particular, when compared with nAMD.[26] The small number of studies in this field have documented anxieties around progression of dry AMD and GA, hopelessness and despair regarding the lack of treatment options, and fears for a future with more limited sight and greater dependence on others.[27–29] A particular concern which participants have raised in qualitative studies is the very limited provision of information or advice at the point of diagnosis, beyond being told that nothing can be done for the condition. The lack of information, or the unclear and inaccessible nature of available information, may compound the sense of confusion and frustration many individuals living with the condition may feel. Therefore, in light of limited research on the impact of GA on patients' quality of life, we will explore patients' understanding and experience of their condition alongside the core theme of treatment acceptability.

We will use semi-structured interviews, designed with patient input to explore themes around patient perspectives on GA progression and current treatment options, the impact of GA on their quality of life, and their perceptions of the acceptability of intravitreal injections and associated burdens. Particular attention will be placed on patients' perceptions of GA treatments which (unlike nAMD treatments) will not reverse or stop vision loss, but only slow down the rate of vision loss.

### Study objectives

Our objectives are to explore the following research questions:
- ► How acceptable are regular eye injections to people with GA?
- ► What are the factors that would affect acceptability of regular eye injections for people with GA?
- ► What is the level of treatment burden that patients with GA will accept in order to slow further visual decline?

**Table 2** The seven component constructs in Sekhon *et al*'s theoretical framework of acceptability (TFA),[25] and how these could apply to intravitreal treatment for GA

| Component construct in TFA | Definition within the TFA | Example with potential relevance to GA treatment |
|---|---|---|
| Affective attitude | How an individual feels about the intervention | Anxiety about the injection, despair and fear of losing vision, or hope of slowing vision loss. |
| Burden | The perceived amount of effort that is required to participate in the intervention | The challenges of monthly visits to clinic for injections, eg, associated pain and discomfort, transport issues, or potential impact on accompanying relatives. |
| Ethicality | The extent to which the intervention has a good fit with an individual's value system | Some individuals with GA may be more proactive and feel they can take control by having injections. Meanwhile, other individuals could be more fatalistic (or accepting) about the inevitability of vision loss, especially if treatment outcomes are unclear or uncertain.[60] Our patient advisors also highlighted that some people with GA may have concerns around the high expense and resource implications for the NHS. |
| Intervention coherence | The extent to which the participant understands the intervention and how it works; the face validity of the intervention for the recipient | Clear understanding of the impact the intravitreal injections would have, in terms of slowing down the rate of vision loss from GA (rather than halting or reversing it). |
| Opportunity costs | The extent to which benefits, profits or values must be given up to engage in the intervention | If a person with GA (and/or an accompanying relative/caregiver) has to take time off work or cancel commitments to attend injections. |
| Perceived effectiveness | The extent to which the intervention is perceived as likely to achieve its purpose | An appreciable sense that the intravitreal injections are slowing the patient's rate of vision loss. |
| Self-efficacy | The participant's confidence that they can perform the behaviour required to participate in the intervention | Confidence in ability to attend regular injections and to persist with treatment over the long term. |

GA, geographic atrophy; NHS, National Health Service.

► How do people with GA understand and perceive their condition, and current and future treatment options?

## METHODS AND ANALYSIS
### Study design
This is a mixed-methods cross-sectional study, which will start in April 2021. At the time of writing, the COVID-19 pandemic and associated restrictions have had profound medical and social repercussions for older adults in particular (the group most at risk of GA). It is important to continue to involve older adults in non-COVID-related research while minimising risk and in line with government guidelines.[30] As such, the decision was made to conduct the research remotely, following discussion with our patient advisory group. Conducting the study by telephone, as opposed to videoconferencing, was considered the most accessible and appropriate medium in order to standardise the procedure and to avoid digital exclusion of certain participants.[31 32]

Telephone interviews will be arranged at a time convenient for participants, and will last approximately 90 min. Interviews will be audio recorded with participants' consent. In addition, participant responses to multiple choice or Likert-type scale questions will be inputted into Qualtrics, with responses delinked from any personal identifiable information.

The interview flow is displayed in figure 2, and the eight interview components are explained in turn below.

### Sociodemographic questions
Participants will be asked basic questions regarding sociodemographic factors, including age, ethnicity, primary language, highest education level and employment status. This section will also ask about participants' living situation, whether a relative or caregiver normally accompanies them to hospital appointments, and time taken and mode of transport to travel to hospital appointments.

### Patient-reported outcome measure: the EQ-5D-5L
We will verbally administer the five level EQ-5D (EQ-5D-5L), which provides a standardised measure of participants' generic health state across five dimensions: mobility, self-care, usual activities, pain/discomfort and anxiety/depression.[33]

### Structured questions about knowledge and concerns regarding GA
We will ask Likert-type scale questions to gauge participants' knowledge about their GA, and how they perceive its severity and likely progression.

i. Socio-demographic questions

ii. EQ-5D-5L

iii. Structured questions about baseline knowledge and concerns regarding GA

iv. Delivery of information about treatment options

v. Structured questions to assess thoughts on the treatment options

vi. Semi-structured questions about treatment options

vii. Discrete choice experiment (DCE)-style task

viii. Broader questions on the impacts of living with GA

**Figure 2** Summary of overall study procedure. GA, geographic atrophy.

## Delivery of information about treatment options

At this point, we will provide participants with brief information about interim findings of clinical trials of GA treatments under investigation (see table 1). The interviewer will explain that treatments have been found to be safe and slow down the rate of progression of GA by up to 30%. A concrete example will be provided in terms of how the injections will help preserve functional vision for up to 30% longer compared with no treatment, since comparisons framed in absolute rather than percentage terms are generally more easily understood.[34 35] As suggested by our patient advisory group, we will also include information on potential risks and burden of treatment, including side effects, pain and discomfort, potential risk of developing nAMD,[36 37] and anxiety that intravitreal injections may cause. Participants will be invited to ask questions and the interviewer will clarify aspects of the information.

## Structured questions to assess thoughts on the treatment options

Participants will be asked brief three-point and five-point Likert-type scale questions to gauge their views on the benefits versus risks of the treatments described. They will be asked if they would be willing to have injections at different time intervals (each month, every 2 months, every 3 months or every 6 months).

## Semi-structured questions about treatment options

Participants will then be asked a series of open-ended questions to explore their thoughts on advantages and disadvantages of the various treatment options, and barriers and facilitators of committing to treatment. While questions can be adapted depending on participants' responses and central concerns, a core set of questions will explore how participants evaluate the probable benefits of treatment against the burden, opportunity costs and risks of treatment. Participants will be invited to discuss how the treatment process would impact not only them but also relatives, friends and caregivers in their life, including those who may accompany them to hospital appointments. We will also ask for participants' views on support and information that could help with decision-making and coping with challenges of treatment. If the participants have experience of intravitreal injections for any indication, they will also be asked to elaborate on their experiences of receiving these.

Participants will be deliberately encouraged to say as much as they can in response to the questions. The interviewers will provide prompts to encourage participants to share their views, while taking special care to avoid saying anything detailed that could be interpreted as leading or coercive.

## Discrete choice experiment (DCE)-style task

In a DCE, participants are presented with two or more alternative products or scenarios, and asked to choose which they believe would be most appropriate. A true DCE would typically be administered to a larger number of participants (based on a sample size calculation), with the attributes of interest (eg, frequency of injection) and various levels (eg, once per month, once every 2 months and so on) computer-randomised to compile different choice sets.[38 39]

In contrast, in our pilot exploratory study, we will ask participants to directly compare four hypothetical scenarios of different treatments and outcomes, presented in pairs in random order, to assess patient preferences for treatment options. One option is a 'No treatment' option, while the other three options are based on the treatment attributes of APL-2, Brimonidine and Zimura.

Based on feedback from our patient advisory group, we are not asking participants to choose between treatment options for themselves, in order to avoid confusion when the option may not be relevant to them (eg, if they themselves would not benefit from treatment due to the advanced stage of their GA). Instead, we will ask them to compare different treatment options for an imaginary patient. We will subsequently ask participants which option they would choose after weighing up advantages and disadvantages. We will acknowledge that they may not necessarily recommend either option, but for the

**Option 2**

Mr Smith goes for an eye injection in clinic every 2 months (6 times per year) for his dry AMD. Each clinic visit takes up to 2 hours.

Severe side effects are rare, but Mr Smith has up to a 1 in 10 chance of developing wet AMD in a year.

Mr Smith will be able to watch the television for 6 more years.

**Option 4**

Mr Smith goes for an eye injection in clinic every 3 months (4 times per year) for his dry AMD. Each clinic visit takes up to 2 hours.

Severe side effects are rare. Mr Smith has up to a 1 in 20 chance of developing wet AMD in a year.

Mr Smith will be able to watch the television for 5 and a half more years.

**Figure 3** Example of two of the options to be compared and discussed in the discrete choice experiment-style task. AMD, age-related macular degeneration.

purposes of this exercise, they should select one of the two options.

Participants will be asked to 'think out loud' and explain their decisions in as much detail as possible. This technique has been used to generate qualitative data from a DCE exploring patient preferences for nAMD treatment[40]; the collection of qualitative data within a DCE task can help detect and resolve participant misunderstandings, as well as provide a more nuanced understanding of how participants' different contexts and values may affect their decision-making.

Often DCEs rely on visuals to clarify the attributes and levels. Pre-piloting with our patient advisory group suggested that these are unlikely to be helpful for a population living with GA and when conducting the interview by telephone. Therefore, we have sought to make the text-based option cards as simple and clear as possible. An example is shown in figure 3.

### Broader questions on the impacts of living with GA

Before concluding the interview, we will use three open-ended, semistructured questions inviting participants to briefly discuss impacts of GA on their everyday life and how these may relate to their hopes for treatment. We will conclude by asking participants how they found the interview experience, and if they wish to discuss any outstanding issues.

### Patient involvement

During the study's conception phase, we convened a group of eight individuals living with GA who will be involved throughout the study's entire life cycle. More specifically, we have conducted two rounds of involvement activities, which have all been conducted remotely (by phone) due to COVID-19 restrictions and have helped refine the study's research question and methodology.

During one-to-one discussions with each of the eight advisory group members, we explored their knowledge and understanding of GA, experiences living with GA and their hopes for what treatment might achieve. These

conversations informed the content and length of the structured and semistructured questionnaires, and the design of the DCE-style task. We also confirmed with them whether they would like to remain engaged in advising us throughout the study in the longer term, including advising on optimal media to disseminate study results.

### Participants

### Recruitment

We will use a purposive sampling strategy, aiming to achieve maximum variation in our sample,[41] in terms of demographic characteristics such as age, gender and ethnicity, as well as in terms of living situation and grade/severity of GA. We aim to recruit 30 individuals with GA, including 15 participants with a history of previous intravitreal injections (for nAMD) and around 15 participants with no experience of intravitreal injections. This will help explore whether and how participants' perspectives may differ depending on their experience of previous intravitreal injections. We will also ensure that at least five participants with fovea-involving GA are included, since this could feasibly influence their perspective on the acceptability of injections to slow down the progression of GA.

Potential participants will be identified from patients with GA who attend NHS Medical Retina clinics at Central Middlesex Hospital, London. Individuals who may be interested and eligible to take part will be approached by their consultant ophthalmologist (author CD) or clinical research fellow (author AG), who will explain the background, aims and nature of the project. Potential participants expressing an interest will then be posted a participant information sheet (PIS) in an accessible format. After an interval of at least 24 hours to read and consider the PIS, authors CD, DJT, AG or JE will contact the potential participant to provide an opportunity to discuss any queries or uncertainties regarding the study. If the participant is willing to proceed, informed verbal consent will be obtained by the relevant author. Consent will be audio recorded, and the relevant author will record this in writing on Qualtrics, with the participant's permission. It will be made clear that even having consented, the participant is free to withdraw from the study at any point without any consequence. Once informed consent is obtained, the interview will take place.

### Eligibility criteria

We seek to include 15 participants with nAMD in their fellow eye and a history of intravitreal injections. For the other 15 participants, it will be required that there is no history of nAMD or intravitreal injections.

Inclusion criteria across all participants will be as follows: age ≥50 years; a diagnosis of GA (bilateral or unilateral); and sufficient understanding of and fluency in English to be able to understand and respond to interview questions.

Individuals will be excluded in case of macular disease in either eye due to causes other than AMD (eg, diabetic

macular oedema, Stargardt disease); any concurrent ocular or intraocular condition that could contribute to central visual impairment; or significant systemic disease or medication known to affect central visual function. We will also exclude participants if they are unable to understand and retain the study information in order to provide informed consent.

## Sample size

This pilot mixed-methods study can be seen as qualitatively driven,[42] which is to say that we envisage the qualitative component as 'a complete study that could be published alone, but it is complemented with another data set'.[43] For this reason, we have not calculated a sample size or prespecified a data saturation point.

In this study, we adopt a realist approach to qualitative research, where we assume that participants' accounts are a straightforward, direct reflection of their actual thoughts and experiences.[44] It is impossible to set a theoretical limit in advance for how many interviews will provide us with new and meaningful information.[45]

Therefore, the sample size of 30 was chosen pragmatically to allow a wide range of perspectives to be represented in the dataset while also being a realistically attainable goal. It will allow us to explore the variation between, as well as complexities within, participants' views on GA treatment acceptability; particularly contrasting those of the ~15 participants with nAMD and a history of intravitreal injections, and the ~15 participants with GA only who are completely naïve to AMD treatment.

## Data analysis (quantitative)

Statistical analysis will largely be descriptive, functioning as a complement to the qualitative data. We will carry out basic, exploratory analysis to explore any correlations between scores on our novel Likert-type questions, EQ-5D scores and sociodemographic variables. However, the small sample size will limit the nature of any preliminary conclusions that can be drawn from the statistical analysis.

## Data analysis (qualitative)

Recordings of interviews will be transcribed verbatim and then analysed using the Framework Method.[46 47] This method of analysis was chosen as it is well suited to applied, multidisciplinary health research with relevance for policy and practice. A key element of the analytical process is generation of a matrix where each row is a 'case' (participant) and each column a 'code' (thematic concepts or issues). Codes may be developed deductively, according to predetermined areas of interest, or inductively, based on open coding of meaningful content in participants' accounts. The matrix provides a practical tool for comprehensive, transparent summarising and analysis of data that facilitates team-working and joint discussion throughout the analytical process. Analysis of categories of codes across the whole matrix allows for the generation of themes that can encapsulate and explain

findings. The qualitative software package NVIVO V.10.2 (QSR International, Cambridge, Massachusetts, USA) will be used to manage the matrix framework.

We will use Consolidated criteria for Reporting Qualitative research (COREQ)[48] as a checklist to encourage transparency and reflexivity when writing up our findings, while mindful of critiques that such checklists do not necessarily reflect best practice in qualitative research.[49]

## ETHICS AND DISSEMINATION
### Ethical considerations

The study received a favourable opinion from the Proportionate Review Subcommittee of the NHS South Central–Berkshire Research Ethics Committee on 10 March 2021 (REC reference: 21/SC/0085). Full ethical approval from the Health Research Authority was granted on 23 March 2021 (IRAS Project ID: 287824).

With participants' consent, telephone interviews will be audio recorded and transcribed. All data will be anonymised, with all personal data, such as names, addresses, dates of birth and so on, being removed. Research data will be stored separately from personal identifiable information. Identification of research data will be possible by investigators by means of a unique alphanumerical identifier for each participant. Only investigators will be able to identify the participants using the alphanumerical identifier.

Anonymised data will be stored electronically for 5 years on encrypted, password-protected NHS Trust storage. These passwords will only allow access by the researchers.

Any personal data that is transferred electronically will be encrypted during transfer. Any hard copies of personal data will be stored within the hospital in locked filing cabinets.

### Dissemination

Participants may opt to receive a summary of the research findings at the conclusion of the study. A document will be compiled with input from our patient advisory group, describing the findings in lay terms. We also hope to disseminate findings to patients with GA (beyond patient advisors and study participants) through talks and lay publications, for example, via support groups run by the Macular Society.

Findings will also be disseminated through peer-reviewed journal publications and conference presentations. We expect significant interest from the medical retina community, and hope that findings will inform therapy development and service delivery planning with regard to the GA treatments under clinical investigation.

## DISCUSSION

Clinical trials of new intravitreal treatments for GA show promising results. However, it is unknown whether people with GA would accept frequent intravitreal injections for

a benefit currently limited to slowing, rather than halting, GA progression.

This study represents a first step towards answering that question, in a specific context of one GA population in London. A key strength of our study is the involvement of a patient advisory group who have helped us to shape the content and design of our research; however, a limitation is that conducting the study remotely by telephone risks missing meaningful non-verbal information, such as facial cues and body language.[50] While this pilot study will not provide a generalisable, definitive answer, we believe that results will provide an initial basis for considering GA treatment acceptability and the determining factors. These barriers and facilitators may be more psychosocial—in terms of hopes or concerns of the patient (and their relatives or caregivers, if relevant)—or more attributable to features of the treatment schedule or the health service, such as frequency of treatment or time spent in clinic. As such, the study will generate useful insights to consider how benefits and risks of treatment can best be communicated to patients, while also considering if certain treatment options and schedules are clearly preferred over others. We also anticipate that the research will generate additional insights into the experience of living with GA, which can inform broader patient support and patient education strategies.

This study will form the basis for a larger quantitative study to assess treatment preferences in GA more empirically across a broader range of participants. There are existing quantitative measures to assess acceptability used in psychosocial research, such as the Treatment Acceptability/Adherence Scale[51] and the Treatment Perception and Preferences measure.[52] The Theoretical Framework for Acceptability, developed by Sekhon and colleagues, is also being used to develop questionnaire items which could be applied generically to assess intervention acceptability.[25] Therefore, our study will help us to consider whether these measures may be appropriate for a larger study, or how we might construct and validate our own questionnaire. To our knowledge, there is no pre-existing, validated measure within ophthalmology with which to assess treatment acceptability. Instead, acceptability tends to be investigated using a combination of qualitative data and Likert-type scale questions (as in this pilot study).[53 54] Therefore, we hope that this pilot and the next phase of research will not only help to understand empirically how patients with GA perceive emerging treatment options, but also to further understanding of treatment acceptability in ophthalmology more broadly.

**Acknowledgements** We would like to thank the eight members of our GA Patient Advisory Group for their time and invaluable input into the study so far. We also wish to thank Jenny Coelho and Bhavna Patel for their help with writing summaries of the discussions between the research team and Patient Advisory Group.

**Contributors** CD, DJT, JE, AG and DPC designed the study and study protocol. CD and DJT acquired funding for the study. JE drafted the manuscript. All authors provided critical comment, revision and feedback on the manuscript and take responsibility for the manuscript content. The corresponding author attests that all listed authors meet authorship criteria and that no others meeting the criteria have been omitted.

**Funding** This work has been supported by the National Institute for Health Research (NIHR) Enabling Involvement Fund (EIF) (Grant number EIFApp ID: 397) and the City, University of London School of Health Sciences Higher Education Innovation Fund (HEIF).

**Competing interests** CD has served on advisory boards for Novartis, Allergan and Apellis. JE, AG and DJT have no interests to declare. DPC reports grants from Roche, grants and personal fees from Santen, grants and personal fees from Apellis, grants from Allergan, personal fees from Thea, personal fees from Bayer and personal fees from Centervue, outside the submitted work. DPC receives funding from the Innovative Medicines Initiative 2 Joint Undertaking under grant 116076 (Macustar). This joint undertaking receives support from the European Union's Horizon 2020 research and innovation program and European Federation of Pharmaceutical Industries and Associations (EFPIA). The communication reflects the author's view and that neither IMI nor the European Union, EFPIA, or any Associated Partners are responsible for any use that may be made of the information contained therein.

**Patient consent for publication** Not required.

**Provenance and peer review** Not commissioned; externally peer reviewed.

**ORCID iDs**
Jamie Enoch http://orcid.org/0000-0002-4614-6676
Arevik Ghulakhszian http://orcid.org/0000-0002-5503-3892
David P Crabb http://orcid.org/0000-0001-8754-3902
Christiana Dinah http://orcid.org/0000-0002-0815-4771
Deanna J Taylor http://orcid.org/0000-0001-8261-5225

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
