## [Reviewer comments · BMJ Open]

ARTICLE DETAILS

TITLE (PROVISIONAL)	Acceptability of intravitreal injections in geographic atrophy: Protocol for a mixed-methods pilot study
AUTHORS	Enoch, Jamie; Ghulakhszian, Arevik; Crabb, David; Dinah, Christiana; Taylor, Deanna

VERSION 1 – REVIEW

REVIEWER	Amoaku, W Univ Nottingham, Ophthalmology
REVIEW RETURNED	15-Feb-2021

GENERAL COMMENTS	This study is well timed as acceptability of intravitreal injections for GA need to be fully explored. A few minor factual corrections are recommended. In addition, it is suggested that it is particularly important to include participants with 'centre-involving GA' in at least one eye. This is vital as the level of VA is dependent on location of the GA, and participant perception and/or acceptability may be dependent on VA (poor VA in one eye might make injections in the other eye more acceptable). As such, at least another 5-10 participants with centre-involving GA in 1 eye should be included in this pilot. Specific comments: Pg 5 line 8: omit 'eye' Pg 10 line 35: insert 'for any indication' after 'injections'(unless limited to particular indications) Pg 10 line 40: insert 'detailed' after 'that' Pg 11 line8: insert 'in order to' preceding 'to avoid' Pg 12 line 20: replace 'geographic atrophy' with 'GA' (abbreviation previously adopted) Pg 12 line 21: change 'to include' to 'including' Pg 12 Line 22: change 'neovascular (wet) AMD' to nAMD (previously adopted) Please use British spelling include 'oedema'
--

REVIEWER	Pardhan, Shahina Anglia Ruskin University, Vision and Eye Research Unit (VERU), School of Medicine
REVIEW RETURNED	11-Mar-2021

GENERAL COMMENTS	This is well thought out protocol for the study and I have no problems with it being published.
---

VERSION 1 – AUTHOR RESPONSE

Reviewer: 1

Dr. W. Amoaku, Univ Nottingham

Comments to the Author:

This study is well timed as acceptability of intravitreal injections for GA need to be fully explored.

A few minor factual corrections are recommended. In addition, it is suggested that it is particularly important to include participants with 'centre-involving GA' in at least one eye. This is vital as the level of VA is dependent on location of the GA, and participant perception and/or acceptability may be dependent on VA (poor VA in one eye might make injections in the other eye more acceptable). As such, at least another 5-10 participants with centre-involving GA in 1 eye should be included in this pilot.

Thank you for your helpful feedback, and in particular this very interesting and useful observation about 'centre-involving GA'. It makes sense that participants with foveal involvement in one eye could have a particular perspective on acceptability of injections for their other eye. We have added a point to this effect, about involving at least 5 participants with centre-involving GA, on page 11 paragraph 2.

Specific comments:

Many thanks for your careful reading and for providing these detailed suggestions. We have made all the changes as you suggested.

Pg 5 line 8: omit 'eye'

Removed as suggested (page 3, paragraph 1).

Pg 10 line 35: insert 'for any indication' after 'injections'(unless limited to particular indications)

Added on page 10, paragraph 3.

Pg 10 line 40: insert 'detailed' after 'that'

Added on page 10, paragraph 4 (although we inserted 'detailed' before rather than after 'that').

Pg 11 line8: insert 'in order to' preceding 'to avoid'

Added on page 11 paragraph 2.

Pg 12 line 20: replace 'geographic atrophy' with 'GA' (abbreviation previously adopted)

Replaced, on page 12 paragraph 2.

Pg 12 line 21: change 'to include' to 'including'

Changed, on page 12 paragraph 2.

Pg 12 Line 22: change 'neovascular (wet) AMD' to nAMD (previously adopted)

Changed, on page 12 paragraph 2.

Please use British spelling include 'oedema'

Changed, on page 13 paragraph 2.

Reviewer: 2

Prof. Shahina Pardhan, Anglia Ruskin University

Comments to the Author:

This is well thought out protocol for the study and I have no problems with it being published.

Thank you very much for your kind feedback.